# Factors Influencing the Use of Geospatial Technology with LiDAR for Road Design: Case of Malaysia

**Fazilah Hatta Antah [1,*], Muhamad Azry Khoiry [1,*], Khairul Nizam Abdul Maulud [1,2]**
**and Ahmad Nazrul Hakimi Ibrahim [1,3]**

1   Department of Civil Engineering, Faculty of Engineering and Built Environment,
    Universiti Kebangsaan Malaysia, Bangi 43600, Selangor, Malaysia; knam@ukm.edu.my (K.N.A.M.);
    nazrulhakimi@ukm.edu.my (A.N.H.I.)
2   Earth Observation Centre, Institute of Climate Change, Universiti Kebangsaan Malaysia,
    Bangi 43600, Selangor, Malaysia
3   Sustainable Urban Transport Research Centre, Faculty of Engineering and Built Environment,
    Universiti Kebangsaan Malaysia, Bangi 43600, Selangor, Malaysia
*   Correspondence: p106635@siswa.ukm.edu.my (F.H.A.); azrykhoiry@ukm.edu.my (M.A.K.)

**Abstract:** This study seeks a better understanding of the acceptance of geospatial technology with Light Detention and Ranging (LIDAR) in road design in a developing country, Malaysia. Existing surveying measurement methods to provide quick, accurate, and reliable information are unsuccessful in producing an expected result, especially in large areas. In addition, topographic data cannot be observed well with the conventional total station method in areas under thick canopies, which is challenging to identify road areas at risk to the environment, such as slope failure. Geospatial surveying technology by LiDAR helps in measuring fields over a wide area and provides a broader spatial extent. At the same time, the laser capability of airborne LiDAR, which penetrates the canopy, helps give accurate readings on the terrain. However, the use of LiDAR geospatial technology for use in road design is still insufficient to date. Thus, this study is developed to identify the factors that influence the use of LiDAR in road design among engineers. Factors identified are barriers, motivation, and strategy. Barrier factors consist of lack of knowledge, risk, cost, and human aspects that slow down the development of LiDAR use. On the other hand, motivational factors consist of encouraging engineers to obtain knowledge about LiDAR and to use it more widely. Meanwhile, a strategy factor form increases LiDAR measurement methods through activities or work procedures. The finding shows that barriers and strategy factors are the significant factors that affect the acceptance of LiDAR among engineers. However, motivational factors have no significant effect to engineers in accepting the use of LiDAR. The advantages of this study and its limitations are also discussed. Finally, this study also provides compilation of few suggestions pertaining this topic to improve future research.

**Keywords:** geospatial technology; LiDAR; sustainable road planning and design; factor influencing road infrastructure project; Exploratory Factor Analysis (EFA); Confirmatory Factor Analysis (CFA)

## 1. Introduction

Geospatial technology acquires, collects, and stores geographic data that contribute to geographic mapping and analysis [1]. Among the examples of geospatial technology are Global Positioning System (GPS), Geographic Information Systems (GIS), spectral imagery, Light Detention and Ranging (LiDAR), and aerial photography are examples of geospatial technology [2–7]. Although geospatial technology is getting more stable, some obstacles are found to hinder its development; including financial [8,9], lack of knowledge [4], lack of experience [10], lack of training [9], data sharing data is not manageable [11], lack of support from institutional [12], and poor communication among organizations [8].

In addition, incentives are able to increase the use of geospatial technology applications such as the government support, [11,13] encouragement for staff to be involved in the form of training and knowledge sharing so that the use of this technology can be utilised fully by the users in improving the quality of work and thus enhancing work ability [14].

In this study, LiDAR is the selected geospatial technology in road design to obtain topographic data due to the weakness of conventional measurement methods involving the disappearance of spot height data during observation data, especially in areas inaccessible to humans, particularly in hilly slopes and dense forests [15]. LiDAR is very potent in providing Digital Terrain Model (DTM) information which is very useful in determining the road network mapping [16,17]. DTM are also useful in providing information for the construction of slope protection in road areas with high slopes [18,19]. Meanwhile, the Digital Elevation Model (DEM) information obtained from LiDAR can be utilised in determining the drainage structure of the road [20].

Previously most of researchers have studied the barrier factors that influence the use of geospatial technology. Reynard [9] and Jozefowicz et al. [10] examined the barrier factor in influencing geospatial technology. In Canada and the UK, the geospatial technology they studied was GIS technology. At the same time, researcher Henrico et al. [13] further describe the factors of behavioural intention in influencing GIS use in South Africa. Barrier factors influencing geospatial data are also shared by researcher Waterman et al. [11] in the UK and researcher Ali et al. [8] in Pakistan. Researcher de Gouw et al. [4] also studied the barrier factors to use geospatial GIS, Global Navigation Satellite System (GNSS), Imagery and LiDAR in New Zealand.

In comparison, motivational factors are reviewed in detail in geospatial web technology by Hennig [14] in Austria, Germany, and Switzerland. Researcher Eilola et al. [12] also studied the barrier factor in using high-resolution remote sensing images in Tanzania. In this paper, a detailed study will be carried out to examine the factors that influence the use of geospatial technology, namely LiDAR in Malaysia. A total of 3 factors were studied, namely the barrier factor, the motivational factor, and the strategy factor.

A questionnaire instrument consists of items pertaining to the literature study of prior research is constructed to examine the factors which impact the use of LiDAR in road design. Exploratory Factor Analysis (EFA) aids in the development of measurement models. The assessment of this study model was then confirmed using Confirmatory Factor Analysis (CFA).

The following is the order of the topics in this research paper: Section 2 explains the literature review on factors influencing usage of geospatial technology. In Section 3, our hypothesis formulation and conceptual framework are described. Section 4 discusses the study's methodology; Section 5 reports the findings; Section 6 is devoted to discussion; and Section 7 presents the work's conclusion.

## 2. Literature Review on Factors Influencing Usage of Geospatial Technology

Previous academics have investigated various aspects that have influenced the use of geospatial technology. Reynard [9] has reviewed past studies on geospatial technologies using the World Wide Web (Geo Web). The results have identified two barrier factors: financial barriers and technological barriers. Financial barriers include budget constraints in collecting a large amount of geospatial data. At the same time, the technological barrier occurs when there is less efficient experience-wise staff in handling GIS efficiently and thus delay the work handling process.

Jozefowicz et al. [10] stated that in the government sector, barriers such as budgets, roles, responsibilities, and regulations restrict the development of big geospatial data. Nevertheless, in the private sector, geospatial data development does benefit giant companies with its data application, such as geospatial data that is stored in cloud facility provided by the company. However, a shortage of experts in GIS, incompatibilities of input data, and incoherent data systems occur in both sectors. There are limitations associated with the use of geospatial data such as LiDAR, spectral and area imagery, receivers and datasets, and

data portals, according to de Gouw [4], due to shortage of knowledgeable staff, insufficient training, and the high acquisition cost of LiDAR.

On the other hand, Hennig [14] stated motivational factors contributes to the use spatial data integrated in web technology. Variants that that contribute to this factor are teamwork spirit among the staffs, a sense of appreciation in the implementation of work, community involvement, cooperation, enjoyment of work, and additional advantages to job roles pertaining to spatial technology.

Henrico et al. [13] explained that the main intention in using free and open-source software such as QGIS in geospatial technology is to improve the quality of work. In addition, it aids in the learning and training process to improve the knowledge of GIS data sharing.

Waterman et al. [11] identified that in geospatial data sharing has also provide obstacle in LiDAR implementation. Several challenges that limit data sharing in the government sector are the lack of resources to handle a broad scope, difficulty in obtaining accurate data in modelling. Government sector also face difficulty to provide risk information to consumers across the private sector. The gap and inaccuracy of data quality between these two sectors tend to affect users' confidence in accepting the data released.

Ali et al. [8] mentioned organizational barriers are identified in geospatial data sharing. These barriers include poor communication among organizations, missing data policy, irresponsibility of stakeholders who are not alert of their actual roles, weak network access in causing difficulty to access data, and budget constraints during implementation.

Eilola et al. [12] also expressed these barriers associated with the use of remote sensing images such as inadequate ICT equipment compatible during data processing and analysis as well as a lack of skilled and efficient employees in using geospatial technology. In addition, citizens lack exposure to geospatial technology, and they still need institutional support for on to use in actual fieldwork. Full support from the government will ensure remote sensing is implemented comprehensively in the planning stage. A summary of these past studies is shown in Table 1 below.

**Table 1.** Factors influencing the implementation of geospatial technology based on previous studies base on previous studies.

| Author(s)/ Years | Country | Geospatial Technology | Instruments | Factors | Findings |
|---|---|---|---|---|---|
| D. Reynard [9] 2018 | • Canada | • GIS | • Review paper | • Barrier | • Inability to use GIS and analyse the findings is due to a lack of expertise.<br>• Insufficient financial resources.<br>• Handling GIS projects take a long time. |
| S. Jozefowicz et al. [10] 2019 | • The UK | • GIS | • Questionnaire | • Barrier | • Shortage of experts in GIS and remote sensing specialists.<br>• Data and systems are incompatible.<br>• Legal issues. |
| S. de Gouw et al. [4] 2020 | • New Zealand | • GIS<br>• GNSS<br>• Imagery<br>• LiDAR | • Questionnaire | • Barrier | • Staff lack of training and knowledge.<br>• Data acquisition costs. |
| S. Hennig [14] 2020 | • Austria<br>• Germany<br>• Switzerland | • Web technology | • Questionnaire<br>• Analysis website<br>• Analysis of web technology | • Motivational | • Practicing and learning.<br>• Assistance and appreciation among the staffs.<br>• Community involvement.<br>• Cooperation and appreciation.<br>• Enjoyment<br>• Job benefits. |
| S. Henrico et al. [13] 2021 | • South Africa | • GIS | • Questionnaire | • Behavioural intention | • Job performance<br>• Learning & training<br>• Others encouragement & self-willingness<br>• Enjoyment<br>• Job benefits |
| L. Waterman et al. [11] 2021 | • The UK | • Sharing geospatial data | • Questionnaire<br>• Interview | • Barrier | • Lack of data quality<br>• Lack of resources |
| A. Ali et al. [8] 2021 | • Pakistan | • Sharing geospatial data | • Questionnaire | • Barrier | • Poor communication among organisations<br>• Missing sharing data policy<br>• Irresponsibility of stakeholders<br>• Weak network access<br>• Budget constraint |
| S. Eilola et al. [12] 2021 | • Tanzania | • High-resolution on remote sensing images | • Questionnaire | • Barrier | • Inadequate ICT equipment<br>• Lack of expert<br>• Less exposure to technology<br>• Lack of institutional support |

## 3. Hypothesis Formulation and Conceptual Framework

### 3.1. Barrier Factor

Barrier is a factor that influences the use of LiDAR in the form of challenges that prevent the use of measurement methods involving information knowledge, risk, cost, and human aspects.

Previous studies found that there are barriers in terms of knowledge among staff. De Gouw et al. [4] stated that the staffs in the organization did not have good knowledge of LiDAR for the reason of being inexperienced and insufficient data exposure to any form of courses and training. Reynard [9] and Kim et al. [21] also investigate difficulties in employing expertise equipped with geospatial technology expertise such as LiDAR. In addition, there is a skill gap and lack of communication between staff who specialize in computer programming and staff who are not competent according to Reynard [9], Jeppesenet al. [22], and Kim et al.

In terms of cost barriers, according to De Gouw et al. [4] and Grohmann et al. [23], LiDAR has high operating costs but it can produce excellent data. According to Kim et al. [21] and Reynard [9], some organizations have constraints in implementing LiDAR applications. Reynard [9], in his study, also stated that another limitation is shortage of high-end computers in LiDAR data processing process.

In addition, Barazzetti et al. [17] stated that LiDAR data could not stand alone because it needs to be integrated with Building Information Models (BIM) to produce as-built drawings of road project. Meanwhile, LiDAR survey observations pose a high risk to airplanes or helicopters passing by in complex areas are expressed by Pellicani et al. [24] and Schumann et al. [25]. They stated that LiDAR measurement observations has limitation during uncertain weather such as overcast and rain. According to Hammond et al. [26], LiDAR data processing will become more complicated if it involves more detailed or broader area of survey.

As an emerging technology LiDAR is prone to rapid changes in software development and persistent with the latest technological updates according to Gargoum and El-Basyouny [27] Based on the Suleymanoglu and Soycan [28] study, the laser scanning capability of the system affects the accuracy of LiDAR measurements in the field. Moreover, LiDAR data can be affected by terrain condition thus make it difficult in filtering point clouds. Therefore, based on a comprehensive literature review, Hypothesis 1 has been proposed.

**Hypothesis 1 (H1).** *The barrier factor positively affects the usage of LiDAR on road design.*

### 3.2. Motivational Factor

Motivational factor is a factor that influences the use of LiDAR in the form of a strong desire to succeed, such as support, information provision and data management.

According to Häggquist and Nilsson [29], management support in providing awareness of the importance of data applications can be a source of motivational development of LiDAR infrastructure development. Meanwhile, De Gouw et al. [4] pinpoint management support is an important element in providing a conducive working environment to produce dedicated and responsible staffs in handling LiDAR data.

Peterson et al. [30] and Cao et al. [31] also reported the management needs support in providing courses or training on processing and analysing LiDAR data. Kim et al. and Kweon et al. [21,32] said that the management provides complete computer software for staff can analyse LiDAR data perfectly.

According to Rose et al. [33], the person appointed for the observation of LiDAR data should have a good experience, whereas, according to Lin et al. [34] and Enwright et al. [35], the appointed person should be able to produce a topographic data map enclosed with extensive information of dense and mountainous forest areas. Therefore, based on a comprehensive literature review, Hypothesis 2 has been proposed.

**Hypothesis 2 (H2).** *The motivational factor positively affects the usage of LiDAR in road design.*

### 3.3. Strategy Factor

A strategy factor is a component that influences the use of LiDAR in the form of a strategy or proposal to increase the use of LiDAR measurement methods through activities or work procedures. In addition to that, strategy approach that can improve the quality of work as well as benefit the department and employees.

According to Aksamitauskas et al. [36], the observation of survey data with LiDAR takes shortened the time to produce result compared to tradional method of using total station. Based on the study of Olafsson & Skov-Petersen [30], impetuous development of LiDAR knowledge is shown in the organization if the expert take into account the needs of the scope of work involved.

In addition, according to Olafsson & Skov-Petersen [30], personal with expertise with the scope of work will assure impetuous development of LiDAR knowledge in the organization.

The presence of competent staff and experts with work experience in using geospatial technology will definitely help other staffs in completing tasks, according to Olafsson & Skov-Petersen [30]. Peterson et al. [30] and Cao et al. [31] also pointed out that the efficiency of skilled staff in analysing LiDAR data can help disseminating knowledge in the workplace.

Another strategy is to share LiDAR data information on various divisions in the agency to help in decision-making in the management of road design projects. Among the divisions involved are bridge and highway divisions which Gargoum et al. [37] and Guo et al. [38] mentioned in their studies.

De Gouw et al. [4] and Morgenroth and Visser [39] said the strategy of education could provide awareness in cultivating knowledge among professional staff in the agency who work in fields that require exploration of the widespread use of geospatial technology to overcome the problem of lack of knowledge.

Free and open-source software or LiDAR data, which are more accessible than the data obtained by both the government and the private sector for work usage, can help create awareness of LiDAR, according to Roussel et al. [40]. Therefore, based on a comprehensive literature review, Hypothesis 3 has been proposed.

**Hypothesis 3 (H3).** *The strategy factor positively affects the usage of LiDAR on road design.*

### 3.4. Proposed Conceptual Framework

Based on the comprehensive literature review stated above, the study conducted is to explore factors influencing the use of LiDAR in road design in Malaysia based on three factors, as shown in Figure 1 below.

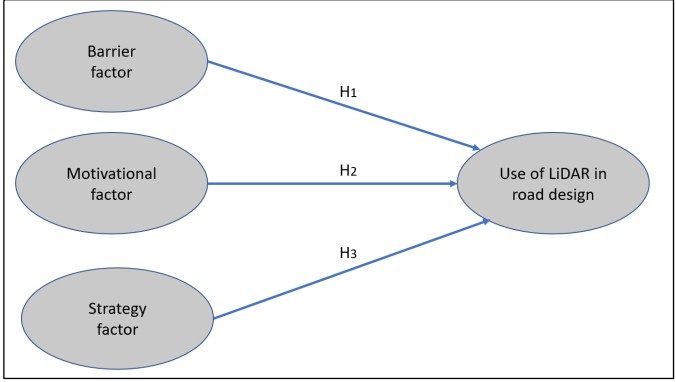

**Figure 1.** Framework of the proposed model.

## 4. Methodology

### 4.1. Instrument Design

First, a comprehensive study was conducted to identify factors that can influence the use of LiDAR in road design in Malaysia by referring to previous studies. Afterwards, a questionnaire was developed by adapting and adopting items from previous studies. After that, the questionnaire was sent for face validation and expert validation [41]. Face validity is essential to ensure each question in the questionnaire prepared is free from grammatical errors and comprehensible [42]. In this study, face validation was conducted after going through comprehensive proofreading. Meanwhile, content validity is crucial to validate the questionnaire developed especially the functionality of each measuring item in the instrument. In this study, an expert panel was appointed to check the reliability of the constructed questionnaire instrument [43].

A pilot study was conducted on 45 samples. The pilot study is crucial to see if the questionnaire is valid and reliable to distribute. Reliability measures the consistency of a constructed instrument, and in this study, the instrument is measured by utilizing with the Cronbach Alpha value [44]. From the results of the pilot study as shown in Table 2 below, the Cronbach alpha values recorded are at excellent level of consistency for the barrier factor and use of LiDAR because it is in the range of coefficients 0.8 to 0.9. In the meantime, strategy factors and motivational factors are at an excellent level because they have an alpha coefficient of more than 0.9. High Cronbach's alpha values, which indicate good internal consistency of the items in the scale [44].

**Table 2.** Reliability of pilot study.

| Factors | Cronbach's Alpha |
| --- | --- |
| Barrier | 0.885 |
| Motivational | 0.952 |
| Strategy | 0.923 |
| Use of LiDAR | 0.853 |

### 4.2. Sample Size and Data Collection

In this study, a simple random sampling method was used. This sampling method is quick and easy [45]. Approximation of the sample size, it was determined with reference to the Krejcie & Morgan [46] table. The population of this study is 300, but the required study sample obtained is 169. In this study, 169 samples were selected because, according to Hair et al. [47], implementing a large sample size will further improve the accuracy of PLS-SEM test results. This number is considered appropriate because it is higher than the minimum sample size obtained through the G*power 3.1.9.7 computation for the sample calculation of a value of 77. In this study, 169 samples were used based on the recommendations of Krejcie & Morgan [46] to represent the population of engineers involved in road design.

As hypothesised in Figure 1 in this research, this questionnaire presents factors and their related measured items in Appendix A, Table A1. The survey questionnaire was distributed in August and September 2021 throughout Kuala Lumpur, Malaysia. Kuala Lumpur was chosen because there is where most of experienced road design work in Malaysia are concentrated. Out of all the questionnaires distributed online, 165 questionnaires were successfully returned by the respondents during data collection. The response rate achieved is 97.6% of the estimated sample size of 169. This rate is outstanding because, according to the Baruch & Holtom [48] study, the response rate for individual studies is around 52.7%.

Of the total, 10 forms were discarded because of missing information. As indicated in Table 3, the questionnaire was constructed on a five-point Likert scale with 155 responses, ranging from strongly disagree with the score one to strongly agree at score five. A total of 59.4% of respondents were males, and 40.6% were females. In terms of education level,

61.9% are degree holders, while 50.3% were equipped 11–20 years of work experience; 57.4% of respondents work in the government sector, and most of them possess high experience using surveying data from total stations/GPS compared to LiDAR and UAV, which is 57.4%.

**Table 3.** Features of the research participants; *n* = 155.

| Characteristics | Items | Numbers | Percentage (%) |
|---|---|---|---|
| Gender | Male | 92 | 59.4 |
| | Female | 63 | 40.6 |
| Educational | Diploma | 12 | 7.7 |
| | Degree | 96 | 61.9 |
| | Master | 44 | 28.4 |
| | Ph.D | 3 | 1.9 |
| Working experience | <1 year | 1 | 6 |
| | 1–10 years | 42 | 27.1 |
| | 11–20 years | 78 | 50.3 |
| | >20 years | 34 | 21.9 |
| Working sector | Government | 89 | 57.4 |
| | Private | 66 | 42.6 |
| Position | Site engineer | 14 | 9 |
| | Design engineer | 63 | 40.6 |
| | Project engineer | 78 | 50.3 |
| Use of surveying data | Total station/GPS | 89 | 57.4 |
| | UAV/Drone | 6 | 3.9 |
| | LiDAR | 20 | 12.9 |
| | Total station/GPS & UAV/Drone | 11 | 7.1 |
| | Total station/GPS & LiDAR | 8 | 5.3 |
| | Total station/GPS, UAV/Drone & LiDAR | 18 | 11.6 |
| | UAV/Drone & LiDAR | 3 | 1.9 |

*4.3. Data Analysis and Tools*

4.3.1. Exploratory Factor Analysis (EFA)

EFA was conducted to determine the item correlation value and the correlation value between the items. In addition, EFA ascertains the number of factors, their interrelationship, and how the variables are related to the factors. [49]. EFA was conducted on 150 samples in this study. This study analyzed EFA analysis using SPSS Statistic 25 software. In EFA, the first steps are Kaiser—Meyer—Olkin (KMO) test for sample size adequacy and the Bartlett sphere test for data factorization [50]. Then it is followed by anti-image correlation matrix [51], commonalities [52] and loading factor [50].

4.3.2. Confirmatory Factor Analysis (CFA): Measurement Model & Structural Model

There are two measurements in the analysis of the actual study. Firstly is measurement the model that contains convergent validity, discriminant validity, and construct reliability. In convergent validity, there is an outer loading values and AVE. Finally, HTMT analysis is incorporated in discriminant validity test whereas composite reliability analysis is included in construct reliability of test.

Variable reliability refers to a study instrument's internal stability and consistency [45]. The reliability test of the variables uses a composite reliability test obtained through the PLS Algorithm procedure. Composite reliability values below 0.6 are considered weak, 0.7 as satisfactory, and 0.8 above are considered as good [45]. The composite reliability value for each variable should be greater than 0.7 [53].

The convergent validity test was determined based on each item's outer loading value and the Average Variance Extracted (AVE) for each variable obtained through the PLS

Algorithm procedure. The AVE value exceeded 0.5, indicating that the study variables explained the average change between the items [53].

In the convergent validity test, outer loading with values above 0.7 indicate that the study items have reached a predetermined level of convergent validity [53]. To solve the problem of items having a factor weighting value less than 0.7, two actions are needed [53]. Firstly, the item of a factor weighting value less than 0.4 should be removed from the analysis. Secondly, for the items of a factor weighting value between 0.4–0.7, testing should be done before decision whether to retain the item or to be removed from the analysis.

The discriminant validity is determined based on the Heterotrait-Monotrait (HTMT) test [47,53] obtained through the PLS Algorithm procedure. The HTMT test ensures that items in one variable are different from items in other variables [53]. Forty-five (45) samples were tested at the pilot study stage for this research. The HTMT test ensures that items in one variable are different from items in other variables [53]. The recommended HTMT value level is less than 0.85 [54] or less than 0.90 [55]. An HTMT value level of less than 0.85 is considered the best because it has a high sensitivity rate to discriminant validity and it shows differences of items in each variable. However, a value level less than 0.9 is still acceptable [47].

The second part of measurement is the structural analysis model, which consists of path coefficient, strength of model R squared (r2), predictive relevant test Q-squared (Q2), and effect size, f squared (f2).

The Path Coefficient is calculated using beta ($\beta$) values, t (t) values, or *p* values [53] from statistical tests through the bootstrapping process. The value is a measure of each independent variable's or predictor's contribution weights in a relationship. The more relevant the predictor variable is determined by the greater the beta value is [47]. When direction of the study's hypothesis is either positive or negative, the value of t defines the level of significance of a relationship, notably $t > 1.28$ ($p < 0.10$), $t > 1.65$ ($p < 0.05$), and $t > 2.33$ ($p < 0.001$). When the study hypothesis had no directional indication, $t > 1.65$ ($p < 0.10$), $t > 1.96$ ($p < 0.05$), and $t > 2.33$ ($p < 0.001$) were used [53].

The model's strength (R squared) aims to assess the degree of changes that occurs to the dependent variable when the independent variable is included in the analysis. The model's strength is measured based on the value of the R squared obtained through the PLS Algorithm procedure. A value of R squared = 0.75 is categorized as strong, 0.50 as moderate, and 0.25 as weak [53]. However, on the other opinion [56], the recommended R squared value is 0.26, categorized as strong, 0.13 as moderate, and 0.02 as weak.

The goal of the predictive relevant test (Q squared) is to determine the independent variable's ability to predict the dependent variable. The blindfolding technique yields Q squared results used to measure relevant prediction. When the Q squared value is larger than 0, the independent variable may be able to predict the dependent variable [53].

The effect size (f squared) determines how much the independent variable contributes to the dependent variable. The effect size uses the PLS Algorithm technique and the value of f squared. A value of f squared = 0.35 was considered high, 0.15 was considered medium, and 0.02 was considered modest [53].

## 5. Results

### 5.1. Exploratory Factor Analysis (EFA)

This study implemented EFA analysis using SPSS Statistic 25 software. There were 30 items tested, as shown in Appendix A, Table A1. By referring to Appendix A-Table A2, the KMO values for barrier factors, motivational factors, strategy factors, and use of LiDAR showed values greater than 0.5 and were accepted [51]. At the same time, the Bartlett sphere test showed a significant account with a reading of <0.05 [51] which was 0.000. Therefore, Bartlett's spherical indicated that all the findings were reliable and met the EFA requirement, while for the anti-image correlation matrix all items higher than 0.5 is acceptable [51]. Then, the analysis proceeded with the Principal Component Analysis (PCA), as shown in Table A2 in Appendix A. The communalities achieved more than

0.5 [52] for all factors indicate that the barrier factor consists of items (BR1 to BR8), the motivational factor consists of items (MV1 to MV9), the strategy factor consists of the item (ST1 to ST8) and use of LiDAR consists of items (ULL1 to UL5).

*5.2. Confirmatory Factor Analysis (CFA)*

5.2.1. Measurement Model

Evaluation of the measurement model was first conducted against the proposed conceptual framework tested in this study through model analysis. The variables found in the conceptual framework are strategy, motivational, barrier, and use of LiDAR. Each variable contains the items as shown in Appendix A, Table A1, From the CFA analysis, 3 factors are found via preliminary testing on 30 items found 3 factors, namely strategy, barrier, and use of LiDAR, with an outer loading value of 0.40 to 0.70. According to Hair et al. [53], items can be eliminated if the AVE value for the factor can be increased. In this study, for the strategy factor, ST8 is eliminated. For the barrier factor, BR1 and BR2 were eliminated, and for the use factor of LiDAR, UL1 and UL 4 were eliminated. Table 4 shows the items eliminated based on the findings of the initial testing of the measurement model conducted.

**Table 4.** Eliminated item.

| Factors | Eliminated Items | Outer Loadings |
|---|---|---|
| Strategy | ST8 | 0.579 |
| Barrier | BR1 | 0.588 |
| | BR2 | 0.597 |
| Use of LiDAR | UL1 | 0.626 |
| | UL4 | 0.458 |

As shown in Appendix A, Table A3 by the second analysis of CFA, all items reached a convergent level of validity because these items meet the outer loading above 0.7 [53].

The AVE values in Appendix A-Table A3 also show that all constructs are above 0.5 [53]. This indicates the constructs is used in this study comply with convergent validity standards.

Table 5 shows the HTMT value of less than 0.85 for each variable studied and, at the same time, shows that the constructs used in this study comply with the standard of discriminant validity [53].

**Table 5.** Discriminant Validity: Heterotrait-Monotrait Ratio (HTMT).

| Factors | Barrier | Strategy | Motivational | Use of LiDAR |
|---|---|---|---|---|
| Barrier | | | | |
| Strategy | 0.583 | | | |
| Motivational | 0.540 | 0.785 | | |
| Use of LiDAR | 0.664 | 0.745 | 0.676 | |

In addition, the composite reliability values for each construct, as shown in Appendix A, Table A3, indicate that the constructs as used in this study comply with the composite reliability [53].

5.2.2. Structural Model

Results of the analysis of the path coefficients test, as shown in and Table 6 below, can answer the hypothesis of this study and the structural model by Smart PLS is shown in Figure 2. The results of hypothesis testing using smartPLS coefficients by bootstrapping technique showed that the barrier factor had a significant relationship with the use of LiDAR because the value of $t > 1.96$ and $p < 0.05$. Then, the strategy factor had a significant relationship with the use of LiDAR significance because the value of $t > 1.96$ and

*p* < 0.05, and the motivational factor had no significant relationship with the use of LiDAR significance because of *t* value < 1.96 and *p* > 0.05 [53].

**Table 6.** Path coefficient.

|  | Path Coefficients (β) | T Statistics | *p* Values | Results |
|---|---|---|---|---|
| H1: Barrier factor->Use of LiDAR | 0.331 | 5.668 | 0.000 | Significant |
| H2: Strategy factor->Use of LiDAR | 0.306 | 2.542 | 0.011 | Significant |
| H3: Motivational factor->Use of LiDAR | 0.182 | 1.587 | 0.113 | Not significant |

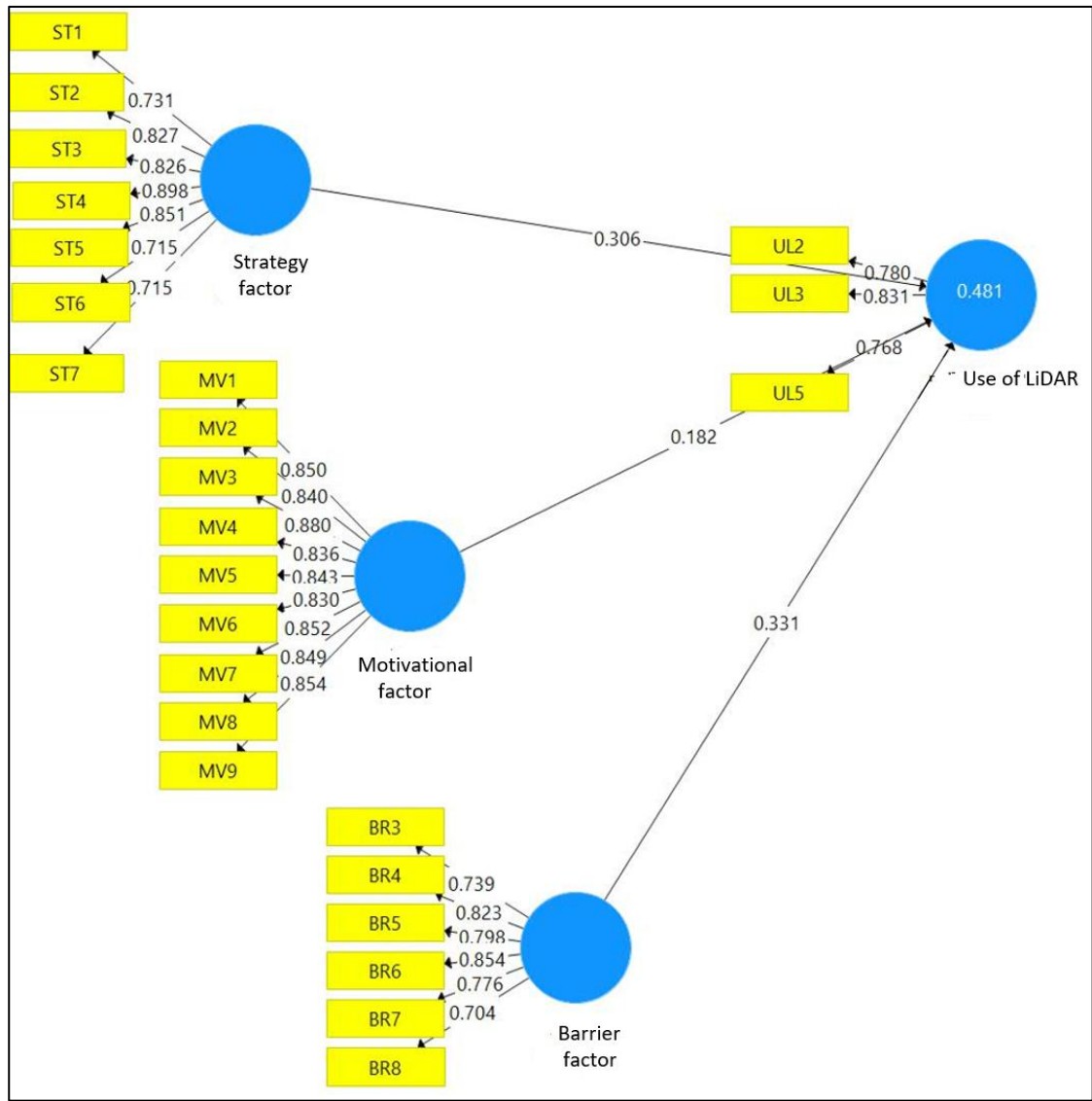

**Figure 2.** Structural model of factors influencing engineers 'acceptance of the use of LiDAR.

Table 7 shows the value of the R squared in PLS algorithm testing and Table 8 is showing the PLS blindfolding test for the value of the Q squared. The value of the R squared obtained is 0.471, indicating the model's strength is moderate [53]. At the same time, the value of Q squared is 0.261, which is greater than zero [53], confirming that the barrier factor, procedure factor, and support factor to predict the relevant impact on the use of LiDAR.

**Table 7.** Results of strength of model test (R squared).

|  | **R Squared** | **Result** |
|---|---|---|
| Use of LiDAR | 0.471 | moderate |

**Table 8.** Results of relevant prediction test (Q squared).

|  | **Q Squared** | **Result** |
|---|---|---|
| Use of LiDAR | 0.261 | relevant impact predictions |

In Table 9, the size effect test, f squared value indicated by the barrier factor is 0.148, strategy factor value is 0.079 and motivational factor value is 0.029 which leads to a small effect size in LiDAR. At the same time, the value of f squared on the motivational factor gives a large effect size [53].

**Table 9.** Results of effect size (f squared).

| **Dependent Variable** | **Independent Variable** | **f Squared** | **Results** |
|---|---|---|---|
| Use of LiDAR | Barrier factor | 0.148 | Small to medium effect |
|  | Strategy factor | 0.079 | Small to medium effect |
|  | Motivational factor | 0.029 | Small to medium effect |

## 6. Discussion of the Results

### 6.1. Theoretical Implications

Despite the apparent benefits of LiDAR for road design work, particularly in forests and highlands, it is still not commonly used in Malaysia. This study aims to focus on the variables that affect the utilization of LiDAR for road design work in Malaysia. Barrier, motivation, and strategy are the aspects examined in this research. Based on the results shown in the previous section, the model developed in this study is supported and sufficiently fit to predict the factors influencing the use of LiDAR in road design work in Malaysia. This study proves that the proposed model leads to 47 per cent of the variance elaborated for the case of intention to use LiDAR for road design among engineers. Furthermore, in this study, all factors are supported except for motivational. Therefore, it is proven that strategy and barrier factors affect the intention to use LiDAR. Interestingly, the most dominant factor influences the use of LiDAR is barrier factor.

The results showed the barrier as the most significant factor in engineers' acceptance of using LiDAR in road design ($\beta = 0.331$, $t = 5.668$). This result is consistent with previous studies by Reynard [9], Ali et al. [8], de Gouw et al. [4], Eilola et al. [12], and Waterman et al. [11].

The studies on barrier factors have shown that limited budgets hampered the use of geospatial technology in most areas, and this result confirmed previous studies carried out by Reynard [9] and Ali et al. [8]. In addition, this study also confirmed the findings by Reynard [9] and de Gouw et al. [4] which explains that barriers also occur due to shortage of staff who are experts in geospatial technology in road design work in Malaysia. This study also proves that inadequate computer equipment to perform high demand data processing of LiDAR have impaired analysis work in road design in Malaysia which resembles the research conducted by Eilola et al. [12] and Waterman et al. [11].

Meanwhile, strategy is the second significant factor in the engineers' acceptance of using LiDAR in road design in Malaysia with a value ($\beta = 0.306$, $t = 2.542$). These results were consistent with studies conducted by Henrico et al. [13] and Schindler [57]. Study by with S. Henrico et al. [13] mentioned that a strategy approach is needed in developing training and learning to maximize the use of geospatial technology.

Schindler et al. [57] conducted a similar study with this research regarding the development of spatial procedure and the importance of considering views of stakeholders

involved in the industry in constructing better guideline. Stakeholder consists of developers, decision-makers, planners, and researchers; and they rely on each other in sharing information, ideas, and data among themselves. Therefore, better efficiently and cooperation between stakeholder can expedite in the usage of geospatial technology.

Interestingly in this research, the motivational factor is not significant, in contrast to the study conducted by Hennig [14], who stated that this factor is crucial because it can drive people to learn and use spatial data to support projects. This research proves only two factors that affect the use of LiDAR in Malaysia: barrier and motivational factors. Therefore, these two factors need to be considered by the Ministry of Works Malaysia, which has significant responsibilities with between departments in Public of Works, which manages the design and construction of both state roads and federal roads. Apart from that, the Malaysian Highway Authority manages the road design of highways in Malaysia.

*6.2. Practical Implications*

Based on this finding, it is hoped that the benefits of geospatial technology offered by LiDAR can be used wisely in road design for further improvements future road design in Malaysia. This study also shows that many engineers involved in road projects agree with the potential usage of LiDAR in road design. It helps to give the info on the volume of cut and fill earthworks in slope areas, map of landslide risk maps in areas which are close to roads despite the limitation pertaining to weather factors such as thick clouds, overcast, and rain.

This study also shows that using LiDAR in road design is beneficial in mapping landslide risk maps such as the landslide risk map in highland areas such as Cameron Highlands, Pahang, Malaysia. It can help engineers design roads by planning the construction of slope protection in areas at risk of slope failure. In addition, the use of LiDAR, which can be used in conjunction with BIM, can assist the design engineers with road networking information between new and existing road using computerised models.

Beside this, other barriers identified are budget for the implementation of LiDAR, handful of LiDAR experts and difficulty in filtering the LiDAR point cloud data due to the density of the earth's surface. Moreover, it may cause shortage of critical information in developing procedures in road design guidelines in Malaysia. To date, there are no specific procedures for road engineers to design roads using LiDAR data and all existing parties are still make use of conventional data from the total station/GPS. Therefore, this study also aids in identifying strategy elements for LiDAR implementation in road design which may give a positive impact in producing guidelines and procedures of road design. The process of producing guidelines of LiDAR should involve person with knowledgeable with relevant subject. The scope of work helps to improve the work process and identifies knowledge and skills needed by the personnel in analysing data.

Barrier identification and proper guidelines implementation in procedures or policy usage of LiDAR in road design will assist policymakers especially Department of Public of Works Malaysia in improvising the existing guidelines or procedures for road design. It will also benefit future road projects by providing in-depth understanding of the flow of road design by using LiDAR data and thus speeding up the road design process particularly in a complex condition. Policy maker should also propose a friendly guideline to ensure LiDAR can be fully integrated in future road projects.

The authorities should implement a promotion programme to raise awareness by expanding a series of training and workshops for all industries and academia to encourage the implementation of LiDAR in road design. Furthermore, to improve the efficacy of the use of LiDAR towards the efficient method of roads and transportation in Malaysia, the Ministry of Works can also expand cooperation between the Ministry of Energy and Resources with organisations like the Department of Survey and Mapping Malaysia.

## 7. Conclusions

The use of LiDAR by engineers in Malaysia was explained using a conceptual model that included the three factors barrier, strategy, and motivation. The concept was proposed and put to the test using SEM. The findings demonstrated how effectively the model predicted engineers' intentions to use LiDAR. Barriers and strategies had a positive impact towards engineers' intentions to use a LiDAR. The most significant factor is a barrier. The barriers elements that were studied in this research were high operating costs, a limited budget to implement LiDAR, a lack of high-end computers, a limited budget to hire expert staff, a limited budget to analyse software, and difficulty in filtering data. This study emphasised the value of LiDAR in road design for engineers and prompted a further study on motivational factors in the future. The findings of this study can aid governments in developing efficient interventions to encourage engineers to use LiDAR. The barrier that restricts the development of LiDAR is shortage of computer equipment and software and it should be taken into consideration by the relevant agencies to expand the use of LiDAR. In addition, guidelines and procedures related to LiDAR should be developed as a reference to industry professionals involved in the design industry for road construction in Malaysia.

**Author Contributions:** Conceptualization, F.H.A. and M.A.K.; methodology, F.H.A. and M.A.K.; validation, F.H.A., M.A.K., K.N.A.M. and A.N.H.I.; formal analysis, F.H.A., M.A.K., K.N.A.M. and A.N.H.I.; writing—original draft preparation, F.H.A.; writing—review and editing, F.H.A., M.A.K., K.N.A.M. and A.N.H.I.; visualization, F.H.A. and M.A.K. supervision, M.A.K., K.N.A.M. and A.N.H.I.; project administration, F.H.A., M.A.K., K.N.A.M. and A.N.H.I. All authors have read and agreed to the published version of the manuscript.

**Funding:** This research was funded by the Minister of Higher Education Malaysia (MOHE), Grant number FRGS/1/2021/TK01/UKM/02/1.

**Institutional Review Board Statement:** Ethical review and approval were waived for this study, due to the study not involving biological human experiment and patient data.

**Informed Consent Statement:** Participants freely decided to participate in the survey and consented to the use of the anonymized data. The need for informed consent statement was waived.

**Data Availability Statement:** All relevant data are within the paper.

**Acknowledgments:** The author would like to thank Universiti Kebangsaan Malaysia (UKM).

**Conflicts of Interest:** The authors declare no conflict of interest.

## Abbreviations

The following abbreviations are used in this manuscript: LiDAR Light Detection and Ranging.

| | |
|---|---|
| TS | Total station |
| UAV | Unmanned Aerial Vehicle |
| DEM | Digital Surface Model |
| DTM | Digital Terrain Model |
| SEM | Structural Equation Model |
| EFA | Exploratory Factor Analysis |
| KMO | Kaiser—Meyer—Olkin |
| CFA | Confirmatory Factor Analysis |
| HTMT | Heterotrait-Monotrait Ratio |
| AVE | Average Variance Extracted |
| CR | Composite Reliability |

# Appendix A

**Table A1.** Measurement items and variables.

| Factors/Items | | References |
|---|---|---|
| BR | Barrier | |
| BR1 | Difficulty in getting expert staff | |
| BR2 | Adequacy of reference to guide | |
| BR3 | High operating cost | Self-created by referring to the research by |
| BR4 | Restricted budget to implement LiDAR technology | Kim et al. [21], S.de Gouw et al. [4], and |
| BR5 | Lack of high-end computers | Grohmann et al. [23], Reynard [9], S. Gargoum and |
| BR6 | Restricted budget on the subscription of paid software to analyse data | El-Basyouny [27], Suleymanoglu and Soycan [28] |
| BR7 | Restricted budget on appointing experts | |
| BR8 | Difficulty in filtering data | |
| MV | Motivational | |
| MV1 | Support from the management is given through exposure to the importance of data application | |
| MV2 | Management support to provide specialized staff | Self-created by referring to the research by Häggquist |
| MV3 | Support from the management is given through providing training | and Nilsson [29], S.de Gouw et al. [4], |
| MV4 | Support from the management is given by providing computer software | Peterson et al. [30], Cao et al. [31], Kweon et al. [32], |
| MV5 | Stakeholders' views are considered in enhancing the knowledge | L. Rose et al. [33]. B.Bigdeli et al. [58], |
| MV6 | Appointed an experienced contractor | B. Babble et al. [59], Z.Zhang et al. [60] |
| MV7 | Appointed a competent contractor | |
| MV8 | Appointed a knowledgeable contractor | |
| MV9 | Providing comprehensive information on dense forest and mountain areas. | |
| ST | Strategy | |
| ST1 | Procedure is developed by those who have the expertise | |
| ST2 | Developed procedure must consider the scope of work | Self-created by referring to the research by Olafsson |
| ST3 | Developed procedure must involve the technical agency | & Skov-Petersen [61], T. Hammond et al. [26], |
| ST4 | Developed procedure must solve problems | A. Shaker et al. [62], S.de Gouw et al. [4], |
| ST5 | Developed procedure must involve an experienced staff | Aksamitauskas et al. [36], of Peterson et al. [30] and |
| ST6 | Procedures developed should identify the knowledge and skills | Cao et al. [31], S. Landry et al. [63] |
| ST7 | Procedures developed should identify the adequacy of training | |
| ST8 | Observation of survey is faster | |
| UL | Use of LiDAR | |
| UL1 | Data obtained detects assets of roads | Self-created by referring to the research by |
| UL2 | Generation of a computerized model | S. Gargoum et al. [37], T. Görüm [64], |
| UL3 | Development of landslide risk maps | P. Jagodnik, et al. [65], F. Hatta Antah et al. [66]. |
| UL4 | Survey data collection is limited to weather factors | B. Matinnia et al. [18] |
| UL5 | The capabilities of accurate data measurement | |

**Table A2.** Analysis of Exploratory Factor (EFA).

| Factors | Code | Kaiser–Meyer–Olkin (KMO) | Bartlett's Test of Sphericity | | | Anti-Image Correlation Matrix of Items | Communalities | Factor Loadings | | | |
|---|---|---|---|---|---|---|---|---|---|---|---|
| | | | Approx. Chi-Squared | df | Sig. | | | Barrier | Motivational | Strategy | Use of LiDAR |
| | | 0.799 | 665.741 | 28 | 0.000 | | | | | | |
| | BR1 | | | | | 0.746 | 0.794 | 0.848 | | | |
| | BR2 | | | | | 0.750 | 0.773 | 0.848 | | | |
| | BR3 | | | | | 0.835 | 0.546 | 0.620 | | | |
| Barrier | BR4 | | | | | 0.828 | 0.742 | 0.839 | | | |
| (BR) | BR5 | | | | | 0.820 | 0.763 | 0.859 | | | |
| | BR6 | | | | | 0.796 | 0.809 | 0.854 | | | |
| | BR7 | | | | | 0.788 | 0.671 | 0.746 | | | |
| | BR8 | | | | | 0.877 | 0.435 | 0.633 | | | |
| | | 0.889 | 1979.794 | 36 | 0.000 | | | | | | |
| | MV1 | | | | | 0.899 | 0.874 | | 0.887 | | |
| | MV2 | | | | | 0.890 | 0.904 | | 0.909 | | |
| | MV3 | | | | | 0.830 | 0.938 | | 0.938 | | |
| Motivational | MV4 | | | | | 0.864 | 0.902 | | 0.908 | | |
| (MV) | MV5 | | | | | 0.974 | 0.808 | | 0.853 | | |
| | MV6 | | | | | 0.881 | 0.941 | | 0.916 | | |
| | MV7 | | | | | 0.837 | 0.956 | | 0.948 | | |
| | MV8 | | | | | 0.918 | 0.931 | | 0.923 | | |
| | MV9 | | | | | 0.838 | 0.878 | | 0.891 | | |

**Table A2.** *Cont.*

| Factors | Code | Kaiser–Meyer–Olkin (KMO) | Bartlett's Test of Sphericity | | | Anti-Image Correlation Matrix of Items | Communalities | Factor Loadings | | | |
| | | | Approx. Chi-Squared | df | Sig. | | | Barrier | Motivational | Strategy | Use of LiDAR |
|---|---|---|---|---|---|---|---|---|---|---|---|
| Strategy (ST) | ST1 | 0.854 | 951.614 | 28 | 0.000 | 0.879 | 0.674 | | | 0.764 | |
| | ST2 | | | | | 0.902 | 0.844 | | | 0.882 | |
| | ST3 | | | | | 0.907 | 0.823 | | | 0.880 | |
| | ST4 | | | | | 0.846 | 0.863 | | | 0.894 | |
| | ST5 | | | | | 0.907 | 0.773 | | | 0.847 | |
| | ST6 | | | | | 0.727 | 0.881 | | | 0.917 | |
| | ST7 | | | | | 0.754 | 0.865 | | | 0.886 | |
| | ST8 | | | | | 0.923 | 0.452 | | | 0.630 | |
| Use of LiDAR (UL) | UL1 | 0.657 | 268.315 | 10 | 0.000 | 0.818 | 0.643 | | | | 0.791 |
| | UL2 | | | | | 0.656 | 0.805 | | | | 0.891 |
| | UL3 | | | | | 0.669 | 0.827 | | | | 0.904 |
| | UL4 | | | | | 0.524 | 0.795 | | | | 0.890 |
| | UL5 | | | | | 0.578 | 0.765 | | | | 0.856 |

**Table A3.** Convergent Validity.

| Factors | Items | Outer Loadings | Average Variance Extracted (AVE) | Composite Reliability (CR) |
|---|---|---|---|---|
| Barrier (BR) | BR3 | 0.739 | | |
| | BR4 | 0.823 | | |
| | BR5 | 0.798 | 0.614 | 0.905 |
| | BR6 | 0.854 | | |
| | BR7 | 0.776 | | |
| | BR8 | 0.704 | | |
| Motivational (MV) | MV1 | 0.851 | | |
| | MV2 | 0.841 | | |
| | MV3 | 0.883 | | |
| | MV4 | 0.839 | | |
| | MV5 | 0.843 | 0.720 | 0.959 |
| | MV6 | 0.827 | | |
| | MV7 | 0.849 | | |
| | MV8 | 0.848 | | |
| | MV9 | 0.851 | | |
| Strategy (ST) | ST1 | 0.729 | | |
| | ST2 | 0.831 | | |
| | ST3 | 0.828 | | |
| | ST4 | 0.900 | 0.636 | 0.924 |
| | ST5 | 0.854 | | |
| | ST6 | 0.709 | | |
| | ST7 | 0.711 | | |
| Use of LiDAR (UL) | UL2 | 0.704 | | |
| | UL3 | 0.754 | 0.629 | 0.836 |
| | UL5 | 0.793 | | |

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
