# Peer review of "Factors Influencing the Use of Geospatial Technology with LiDAR for Road Design: Case of Malaysia"

_sustainability, doi:10.3390/su14158977_

Round 1

Reviewer 1 Report

Briefly describe the three factors in the abstract not necessarily how they were obtained or assessed in the questionnaire: barriers, motivational and strategy.

Line 58 is about conventional surveying not LIDAR. Clarify.

Lime 66: Remove “the” before “researchers”.

Line 67: remove point after reference cutting sentence in two.

Line 107: Sentence unbalanced.

Line 109: Add “the” before “private sector”.

Lines 113-115: Review sentence.

Line 276: Review sentence.

Line 518: Review sentence.

Line 276: Review sentence.  

Line 296: Explain why.

Line 348: Review sentence.

Line 440: Table should be in annex. Also Table on line 491.

Line 518: Review sentence.

Author Response

Thank you Prof/Dr./Sir/ Madam for your prior comment.

Reviewer 2 Report

The paper is definitely a continuation of previous work of the authors on the perceived usefulness of Lidar. In this work, they try to explore factors that affect the use of Lidar in road design.

It is notorious that in the introduction and part of chapter 3 the authors try to sum up parts of previous work. However, it must be better worked and the object of the present work clearly introduced and justified. Only nearly the end of the work the strategic factors are referred to as the contribution of the work to the state of the art. In fact, I questioned several times during my review the usefulness of the model, since most of the information was collected from the literature. Therefore, chapter 3.1.3 must be greatly improved.

In terms of methodology, it follows the main steps.

Tue questionnaire was not presented. However, it should be presented as an annexe or as supplementary material.  

Figure 3 is not essential, it may be removed.

Most tables must be redesigned. They are too big, thus har to read.

Check the content of line 506.

The discussion is very short. The results must be better explored. For example, the items related to “use of Lidar” were not explored.

Chapter 7 is hard to follow. Figure 5 is the last issue presented. I see it as a result of the work. It should be framed, presented and its significance highlighted.

The document requires language and reference editing review. There are incomplete/cut sentences, repeated expressions, stops misplaced, and spaces missing, ….. Also, review acronyms description (particularly chapter 4.3). Expression as thick forest (dense forest), forecast (predict), allocation (budget), theodolite (total station), …. must be checked and changed if appropriate.

Check particularly lines 113-115. For me, the sentence does not make sense.

Author Response

Thank you Prof/Dr. /Sir/ Madam for your prior comment.

Reviewer 3 Report

The paper has good introduction and a good structure in the presentation of the work but needs to be reviewed in order to guide the reader in a good understanding of the point clouds. I strongly recommend a deep review that will give a value to the obtained results through discussion. The results need to be discussed as well, where the need of a discussion section/paragraph. Methodology and data used should be written separately and proper. Result should be expressed in detail with proper explanation. It seems to me that the results are not proportional to the methodological effort made

Author Response

(The authors gave the same response as above.)

Round 2

Reviewer 3 Report

Thank you for accepting my comments.